# Influence of Soluble Guanylate Cyclase on Cardiac, Vascular, and Renal Structure and Function: A Physiopathological Insight

**DOI:** 10.3390/ijms26104550

**Published:** 2025-05-09

**Authors:** Daniele De Feo, Francesco Massari, Cosimo Campanella, Anna Livrieri, Marco Matteo Ciccone, Pasquale Caldarola, Micaela De Palo, Pietro Scicchitano

**Affiliations:** 1Cardiology Section, University of Bari, 70121 Bari, Italy; daniele.df93@gmail.com (D.D.F.); anna.livrieri693@gmail.com (A.L.); marcomatteo.ciccone@uniba.it (M.M.C.); 2Cardiology Section, Hospital “F. Perinei” Altamura (BA), 70022 Altamura, Italy; franco_massari@libero.it; 3Cardiology Department, Hospital “San Paolo” Bari, 70132 Bari, Italy; cosimocamp@gmail.com (C.C.); pascald1506@gmail.com (P.C.); 4Cardiac Surgery Section, University of Bari, 70121 Bari, Italy; micaela.depalo85@gmail.com

**Keywords:** guanylate cyclase, cyclic guanosine monophosphate, endothelial function, renal function, cardiac muscular cells

## Abstract

The role of nitric oxide (NO), soluble guanylate cyclase (sGC), and the cyclic guanosine monophosphate (cGMP) pathway in cardiovascular and renal health and disease is a complex issue. The impact of these biochemical pathways on the vascular tree is well established: the activation of sGC by NO promotes vasodilation and modulates vascular tone. Indeed, additional characteristics exist that lead physicians to believe there is a pleiotropic influence of this pathway on the functional activities and structural characteristics of human tissues and cells. Recently, sGC stimulators have demonstrated clinical efficacy in patients with worsening heart failure with reduced ejection fraction, improving cardiovascular death risk, re-hospitalization for HF, and all-cause mortality. These new outcome data have increased interest in understanding the potential pathophysiological mechanisms. The NO-sGC-cGMP axis may influence endothelial function, kidney performance, and cardiac muscle cell activity. The synergy of these actions could explain the positive effects of vericiguat on worsening HF. The aim of this narrative review was to provide a comprehensive insight into the pathophysiological mechanisms of action of NO-sGC-cGMP axis stimulators on cardiac muscle, endothelial cells, and kidneys.

## 1. Introduction

The role of nitric oxide (NO), soluble guanylate cyclase (sGC), and cyclic guanosine monophosphate (cGMP) pathways in cardiovascular and renal health and disease remains a challenging topic.

In 1998, Robert F. Furchgott, Louis J. Ignarro, and Ferid Murad were awarded the Nobel Prize in Physiology or Medicine for their discovery of nitric oxide as “a signaling molecule in the cardiovascular system” [1]. The biochemical pathways activated by NO were found to regulate vascular tone, while further research identified the NO-sGC-cGMP axis as a key modulator of atherosclerosis-related mechanisms, including smooth muscle cell proliferation, leukocyte migration from the bloodstream to the intima–media space, and platelet aggregation [2].

Alterations in NO production and bioavailability [3,4], sGC activation and stimulation [5,6], and cGMP signaling [7] have been implicated in the pathogenesis of atherosclerosis, heart failure, and kidney disease.

However, studies investigating the impact of nitrates/NO donors or phosphodiesterase V inhibitors—compounds that counteract cGMP degradation—have yielded inconclusive results [8,9,10,11]. Notably, the VerICiguaT GlObal Study in Subjects With Heart Failure With Reduced EjectIon FrAction (VICTORIA) trial [12] recently demonstrated a 10% reduction in the composite risk of all-cause mortality or hospitalization for heart failure (HF) with Vericiguat, a soluble guanylate cyclase stimulator that enhances NO-induced sGC activation, in patients recovering from worsening HF. Similarly, sGC stimulators have been shown to impact kidney function and consistently protect the glomerulus from degeneration [13,14].

The precise mechanisms underlying these effects remain to be fully elucidated. The novelty of this narrative review was to provide a comprehensive insight into the pathophysiological role of the NO-sGC-cGMP pathway in cardiomyocytes, endothelial function, and kidney function. This would improve the current knowledge about novel drugs, which have been recently adopted for the treatment of patients with worsening heart failure, and provide suggestions for possible pleiotropic effects of these compounds.

## 2. Cardiac Muscle Cells

Guanylate cyclase (GC) and intracellular cGMP are key modulators of cardiac cells [15,16]. sGC and membrane-bound (particulate, pGC) guanylate cyclases—such as natriuretic peptide (NP) receptors A and B—are activated by NO and NPs, respectively, thereby influencing endothelial function in the coronary microcirculation, cell growth and apoptosis, myocardial fibrosis, and pathological myocardial remodeling [15,16].

In cardiac cells, two main pathways converge on cGMP activation: the NO-sGC-cGMP and NP-pGC-cGMP pathways [17]. The interplay between these pathways is crucial for cardiac cell regulation, although the precise mechanisms remain under debate.

Vericiguat specifically enhances sGC activity, thus targeting the NO-sGC-cGMP pathway, whereas other pharmacological agents for HF, such as sacubitril/valsartan, primarily act on the NP-pGC-cGMP pathway [17].

Increased cGMP levels influence cardiac cell activity and structural integrity via cyclic nucleotide-gated cation channels (CNGs), phosphodiesterases (PDEs), and protein kinases (PKs) (Figure 1) [16].

The main effectors of cGMP signaling are cGMP-dependent protein kinases (PKG) [18]. Two primary isoforms of PKG have been identified: PKGI, which is further subdivided into PKGIα and PKGIβ via alternative splicing of the Prkg1 gene, and PKGII [19,20]. While PKGII is minimally expressed in cardiac cells, PKGI is widely distributed in cardiac myocytes, cardiac myofibroblasts, endothelial cells, vascular smooth muscle cells (VSMCs), platelets, nerves, and neurons [19,20]. Additionally, PKGI is present in renal cells, leukocytes, and specific areas of the nervous system. Conversely, PKGII activity has been observed in renal cells, the gastrointestinal tract, the pancreas, the parotid and submandibular glands, and certain brain nuclei [19,20].

Although the precise roles of PKGIα and PKGIβ in cardiac cells remain under investigation, evidence suggests that these enzymes interact with various effectors that influence cardiac dynamics and morphology. This interaction may help explain the beneficial effects of drugs such as Vericiguat on cardiac cells, although further studies are needed to address this issue.

### 2.1. cGMP/PKG and the Cardiac Sarcomere

The cGMP/PKG axis significantly influences cardiac sarcomere function, promoting muscle relaxation and enhancing diastolic performance [20]. Preclinical studies have demonstrated the role of cGMP/PKG in modulating key sarcomeric proteins such as cardiac myosin-binding protein C (cMyBP-C), troponin I (TnI), and titin. Emerging evidence also suggests potential interactions with cysteine-rich LIM-only protein 4 (CRP4), troponin T (TnT), and vasodilator-stimulated phosphoprotein (VASP) [21,22,23].

Specifically, PKG-mediated phosphorylation of titin appears to modulate myocardial distensibility and passive stiffness. PKG phosphorylates in vitro sarcomeric titin, reducing its stiffness and improving diastolic function [24]. This effect is further enhanced by direct phosphorylation of TnI, which increases resting cell length, reduces contraction amplitude, accelerates relaxation, and decreases calcium ion (Ca^2+^) sensitivity [25]. While this mechanism induces negative inotropic effects due to reduced myofilament responsiveness to Ca^2+^, it concurrently improves relaxation and diastolic function [26].

cMyBP-C is also a substrate for PKG and is implicated in cardiac (dys) function [27,28,29]. As a key sarcomeric regulator, cMyBP-C plays a critical role in cardiac contractility [29]. In healthy cardiac cells, cMyBP-C is normally phosphorylated; however, HF is associated with reduced cMyBP-C phosphorylation [30,31]. Thoonen et al. [27] demonstrated that PKGα-mediated phosphorylation of cMyBP-C at Ser-273, Ser-282, and Ser-302 in mice subjected to left ventricular pressure overload exerted anti-remodeling effects by enhancing myocardial relaxation.

CRP4, a protein localized at the Z-band of the sarcomere, mediates α-actinin interactions and is implicated in a rare form of dilated cardiomyopathy when altered. Similarly, VASP, which regulates the interaction between cytoplasmic actin and the extrasarcomeric cytoskeleton, influences cardiac cell mechanics. Both proteins are potential PKG substrates; however, definitive data are lacking, and further research is required to confirm their roles [20].

Indeed, cardiac cell function and dysfunction is also related to mitochondria integrity [32]. It is well established that stress conditions—such as ischemia–reperfusion (I/R)—might impair mitochondria activities. I/R might provoke oxidative stress, alter the function of the mitochondrial permeability pore (mPTP), and impair mitochondrial fission and fusion, thus favoring cell apoptotic processes and cardiac cell death [32]. The role of the NO/sGC/cGMP axis in this context is still under debate. PKG seemed to increase the probability of Mitochondrial ATP-Sensitive Potassium Channel (mitoKATP) [33], thus preventing damages to cardiac cells in I/R conditions. It is supposed that the opening of mitoKATP might promote the influx of potassium into the mitochondria, which in turn triggers moderate reactive oxygen species (ROS) production [33]. The mild increase in mitoKATP-mediated ROS production was supposed to activate biochemical pathways, which included the activation of protein kinase C epsilon and the prevention of the mPTP, i.e., one of the main effectors of cell death [34]. PKG might also activate a further potassium channel—the Mitochondrial Ca^2+^-Activated K^+^ Channels of Big Conductance (mitoBK) channels—which seemed to influence mitochondrial energy production and regulation of ROS balance [35].

### 2.2. cGMP and Ca^2+^ Transport

Beyond regulating contractile function and relaxation in cardiac cells, cGMP also mediates Ca^2+^ influx/efflux in the cytoplasm and intracellular calcium storage structures [20]. Preclinical studies [36,37] have shown that the NO-sGC-cGMP axis inhibits L-type Ca^2+^ channels (LTCCs). PKGI effectively suppresses the activity of the voltage-dependent Ca^2+^ channel Ca(v)1.2, which is critical for excitation–contraction coupling [36,37]. Specifically, PKGI-mediated phosphorylation of Ca(v)1.2 increases its closed-state duration, counteracting the phosphorylation effects of adenosine monophosphate (AMP)-activated protein kinase (AMPK) [38]. This prevents excessive intracellular Ca^2+^ ([Ca^2+^]i) accumulation, thereby balancing AMPK-induced LTCC activation and avoiding Ca^2+^ overload.

Additionally, reduced Ca^2+^ overload due to PKGI activity is linked to phosphorylation of the cardiac ryanodine receptor-2 (RyR2) and its regulation of sarcoplasmic reticulum (SR) Ca^2+^ handling [39]. Phosphorylation of RyR2 via AMPK has been shown to improve Ca^2+^ release synchronization, wave propagation, and RyR2 functional adaptation to SR [Ca^2+^] fluctuations [40,41]. In vitro studies [39,42,43] indicate that PKGI-mediated phosphorylation of RyR2 and phospholamban enhances Ca^2+^ uptake into the SR, contributing to reduced [Ca^2+^]i.

PKGI also influences the activity of the sodium/hydrogen exchanger-1 (NHE-1) and the sodium/calcium exchanger (NCX) [44,45,46]. Specifically, PKGI inhibits protein phosphatase 1 (PP1), which otherwise dephosphorylates NHE-1, thereby blocking its ion exchange function [20,46]. This inhibition has significant consequences for cardiac cells: increased intracellular hydrogen ions concentration ([H^+^]i) due to NHE-1 suppression may confer post-ischemic cardioprotection by reducing mitochondrial permeability transition, calpain activation, and hypercontractility [20,47]. However, this effect appears to be NO concentration-dependent: normal NO levels continue to stimulate NHE-1 activity, leading to sodium ion (Na^+^)-driven Ca^2+^ loading, which enhances cardiac contraction (positive inotropic effect) but also predisposes the patient to ectopic beats and arrhythmias [44].

One of the involved pathways related to Ca^2+^ influx in cardiomyocytes is related to transient receptor potential canonical (TRPC)-3 and -6 channels. TRCP-3 and -6 are also modulated by PKG activation [20]. PKGI-mediated phosphorylation of TRPC-3 and TRPC-6 leads to their inactivation [48]. Domes et al. [48] reported that PKGI action on TRPC-3 and -6 reduced cardiac hypertrophy and fibrosis in male mice. Inhibiting these channels decreases [Ca^2+^]i, thereby suppressing the activation of the calcineurin (CN)/nuclear factor of activated T cells (NFAT) pathway, which is responsible for the nuclear transcription of genes promoting cardiac fibrosis and hypertrophy [49,50].

Other mechanisms are also involved in the pathogenesis of cardiac fibrosis and hypertrophy, as the cardiomyocyte-specific deletion of PKGI does not necessarily lead to cardiac hypertrophy [51]. Nonetheless, PKGI activation appears to play a protective role in adverse cardiac remodeling.

### 2.3. cGMP and Cardiac Gene Expression

Understanding the impact of the NO-sGC-cGMP axis on cardiac gene expression through PKGI activity is crucial for evaluating the effects of pharmacological interventions such as Vericiguat on myocardial structure and function.

CN plays a pivotal role in this process, extending beyond its known function in NFAT activation. Fiedler et al. [50] showed that CN activates protein kinase C, c-Jun NH2-terminal kinase, and the transcription factor Myocyte Enhancer Factor 2 (MEF2) in cardiomyocytes, thereby modulating genes associated with hypertrophy. PKGI inhibits MEF2, potentially preventing hypertrophic gene expression independently of NFAT signaling [50].

In another study, Fiedler et al. [52] demonstrated that NO and its second messenger, cGMP, protect cardiomyocytes from apoptosis during ischemia–reperfusion. cGMP activates PKGI, which interferes with the TAB1/p38 mitogen-activated protein kinase (p38 MAPK) pathway, a key regulator of apoptosis [52]. Ischemia–reperfusion might promote apoptosis via several mechanisms. One of them is related to the activation of TAB1, which in turn promotes the phosphorylation and activation of the p38 MAP and the apoptotic processes. The experimental data from Fielder et al. [52] effectively demonstrated that cGMP-activated PKGI inhibits p38 MAPK phosphorylation during simulated ischemia/reperfusion condition via by interacting with its N-terminal Leucine-Isoleucine Zipper. This interaction does not alter the mitogen-activated protein kinase kinase-3 (MKK3) and MKK6 pathways but rather the interaction between TAB1 and p38 MPAK, thus reducing the probability of activating pro-apoptotic biochemical pathways.

PKGI also phosphorylates tuberin (TSC2), a GTPase-activating protein that regulates the mechanistic target of rapamycin complex-1 (mTORC1) [53]. mTORC1 governs cellular growth, metabolism, protein synthesis, and autophagy in cardiac cells via Ras homolog enriched in brain (RHEB) protein [53]. PKGI-mediated phosphorylation of TSC2 inhibits mTORC1, thereby reducing hypertrophic stimuli and promoting autophagy [53].

Moreover, in vitro studies suggest that PKGIα activation influences GATA Binding Protein 4 (GATA4), a transcription factor crucial for human cardiac muscle development, with structural and functional alterations linked to cardiomyopathies [54]. Maintaining the integrity of the PKGI pathway is therefore essential for preserving cardiac cell homeostasis and function.

### 2.4. Inflammatory Regulation

NO plays a central role in modulating cardiac tissue inflammation. Kalra et al. [55] reported that NO induces tumor necrosis factor-alpha (TNF-α) activation, leading to nuclear factor kappa B (NF-κB) transcription and the initiation of inflammatory responses. Specifically, PKGI activates IκB kinase (IKK), which phosphorylates NF-κB subunits p65, p49 (p52), and p50, promoting their nuclear translocation and transcriptional activity [56].

Meanwhile, IKK also activates NF-κB inhibitor IκBα, which binds the p50/p65 NF-κB heterodimer in the cytoplasm, thereby suppressing inflammatory gene transcription [57]. Consequently, the NO/cGMP/PKGI pathway exhibits dual regulatory effects on cardiac inflammation, but the precise mechanisms remain under investigation.

### 2.5. Future Perspective

Although the role of the cGMP/PKG pathway in cardiac function is increasingly recognized, several aspects remain to be clarified. Future studies should better define how the PKG-mediated phosphorylation of sarcomeric proteins like cMyBP-C, titin, CRP4, and VASP impacts myocardial mechanics and disease progression.

The interaction between the cGMP/PKG axis and mitochondrial channels (mitoKATP and mitoBK) also warrants deeper investigation to understand their role in cardioprotection against ischemia–reperfusion injury. Similarly, the impact of cGMP/PKG signaling on cardiac gene expression and hypertrophic pathways shall also be analyzed in the next future.

Finally, clarifying the dual effects of the NO-cGMP-PKG system on cardiac inflammation will be crucial to harness its therapeutic potential in heart failure and other cardiovascular diseases.

## 3. Peripheral Endothelial Function

The systemic increase in sGC activation through the administration of sGC stimulators may contribute to peripheral vascular modifications and impact endothelial function (Figure 2).

HF is well known to be associated with impairments in peripheral vascular function, as demonstrated in clinical studies [58,59]. However, the mechanisms through which sGC stimulators ameliorate endothelial function and vascular wall morphology remain under investigation.

The activation of GC via NO and/or NPs typically promotes the production of cGMP, which in turn regulates vascular tone, cell growth and differentiation, vascular wall permeability to bloodstream cells, and the functionality of platelets and erythrocytes [60].

Thus, the activation of the NO-sGC-cGMP biochemical pathway by sGC stimulators may theoretically contribute to the improvement of vascular function in patients with HF.

### 3.1. Effects of NO-sGC-cGMP on VSMCs

The NO-sGC-cGMP biochemical pathway influences the activity of VSMCs. VSMC proliferation is a key factor in the development and progression of atherosclerosis [61,62]. Specifically, the Ca^2+^/CN axis plays a role in this process: increased intracellular Ca^2+^ promotes the activation of CN, leading to the activation of transcription factors that drive VSMC proliferation through the NFAT pathway [63].

Increased NO levels inhibit intracellular Ca^2+^ concentration by reducing its release from the SR [64,65], while cGMP modulates myofibrillar sensitivity to Ca^2+^ and enhances its uptake by the SR [66]. Consequently, increased cGMP and GC activity may theoretically reduce VSMC proliferation. Li et al. [67] demonstrated that the NO/PKG axis inhibits VSMC proliferation without significantly affecting VSMC viability.

PKGI, derived from the NO/cGMP/PKG axis, plays a pivotal role in VSMC activity and vascular diseases [21]. For instance, PKGIβ phosphorylates the inositol 1,4,5-trisphosphate (IP3) receptor-associated PKGI substrate (IRAG), which inhibits Ca^2+^ release from the SR in VSMCs via IP3 [21]. Furthermore, PKG enhances the activity of large-conductance Ca^2+^-activated K^+^ (BKCa) channels, leading to membrane hyperpolarization and the closure of voltage-dependent Ca^2+^ channels, thereby reducing Ca^2+^ influx [21].

Preclinical studies [21] have further explored PKGI’s role in modulating VSMC contraction and relaxation. PKGI appears to enhance the activity of regulator of G protein signaling-2 (RGS2), inhibiting hormone receptor-triggered Ca^2+^ release and vasoconstriction in VSMCs [68,69]. Additionally, PKGI increases the activity of Ca^2+^/calmodulin-dependent myosin light chain phosphatase by modulating myosin phosphatase targeting subunit-1, ultimately inhibiting myosin light chain phosphorylation and promoting VSMC relaxation [70,71].

A recent study by Schmid et al. [72] investigated the relationship between the NO/cGMP/PKG axis and potassium channel-mediated vasodilation. They found that treatment with NO markedly reduced BK channel activity in VSMCs from rat and mouse arteries [72]. NO might activate BK channels via PKGI, contributing to reduced Ca^2+^ influx and promoting vasodilation [72,73]. Indeed, the literature is not univocally in line with these findings [74,75,76], thus remarking that BKCa involvement in NO-induced vasodilation may be vessel- and context-dependent.

Further research is required to confirm preclinical findings and to elucidate the potential role of Vericiguat in this context, as preliminary data suggest promising effects [77].

### 3.2. Effects of NO-sGC-cGMP on Endothelial Cells

The NO-sGC-cGMP biochemical pathway plays a fundamental role in maintaining proper vascular function and preventing atherosclerosis. Specifically, cGMP may indirectly enhance nitric oxide synthase (NOS) activity in endothelial cells through a feedback mechanism.

While previous preclinical studies [78] have explored the anti-atherosclerotic properties of sGC stimulators, the exact molecular mechanisms remain under investigation.

Hypotheses suggest that cGMP-mediated reduction in TGFβ activity may contribute to decreased fibrosis, proliferation, inflammation, and collagen deposition in the vascular wall matrix [5].

It is well established that endothelial cell-derived NO activates GC in VSMCs, leading to vascular relaxation [79]. Martinelli et al. [80] described a positive feedback loop in which sGC stimulation enhances NO production in isolated endothelial cells, an effect mediated by cGMP-induced eNOS activation [80].

Notably, sGC stimulation supports vascular integrity by promoting angiogenesis and neovascularization in response to ischemic events [81]. Dhari et al. [81] demonstrated that the sGC/PKGI pathway activates p44/p42 MAP kinase, enhancing bone marrow-derived pro-angiogenic cell function and facilitating angiogenesis and neovascularization.

Additional research has highlighted the impact of the NO/sGC/PKGI axis on coagulation balance. Analyses on BAY 41-2272 and BAY 58-2667 addressed this issue. Specifically, BAY 41-2272 is an activator of soluble guanylate cyclase, which weakly binds on the surface of the H-NOX domain, thus stimulating sGC through a non-NO mechanism, being potentiated by the presence of NO [82]. Indeed, BAY 58-2667—also named Cinaciguat—interacts with the heme cavity on the sGC α1 subunit, thus directly activating soluble guanylyl cyclase even in the absence of the heme [82]. The effects of BAY 58-2667 are even more potent than those derived from the action of BAY 41-2272.

Sovershaev et al. [83] reported that both of them reduce tissue factor expression in lipopolysaccharide-stimulated monocytes and TNF-α-stimulated HUVECs, as well as NF-kB transcriptional activity. These findings suggest that they may mitigate pro-coagulative conditions and vascular inflammation, the latter more strongly than the former [83].

Differences between sGC stimulators and activators mainly account for these findings. The action of the stimulators—such as Riociguat and Vericiguat—depends on the oxidation state of the heme binding site on the β subunit of the enzyme, as they need the presence of the ferrous-reduced heme to finally chase their duty of activating sGC [84]. Their action is in parallel with the presence of NO, thus potentiating and emphasizing the activity of NO [84]. Activators, such as Cinaciguat and Runcaciguat, mainly activate the oxidized, heme-free form of sGC, while demonstrating poor influence on the reduced heme enzyme [84]. This is an interesting insight as the preservation of NO reserves might be fundamental for the activity of stimulators, while conditions characterized by oxidative stress and poor provisions in NO might be the perfect field of action of activators. Further clinical analyses are needed in order to better define such an issue.

Finally, thrombin appears to suppress cGMP production, potentially via protease-activated receptor-1 activation, thereby influencing coagulation pathways [85].

### 3.3. Effects of NO-sGC-cGMP on Bloodstream Cells

sGC stimulation affects bloodstream cells, regulating their function. The role of sGC in red blood cells (RBCs) remains a subject of investigation. Cortese-Krott et al. [86] detected sGC activity in human and murine RBCs, finding that its stimulation was independent of NO presence and persisted under conditions of endothelial dysfunction or NO deficiency, such as sickle cell anemia, pulmonary hypertension, and heart failure [86]. Recent studies [87,88] suggest that hypoxia may activate sGC in RBCs, leading to cGMP release and subsequent vascular wall signaling, ultimately providing cardioprotective effects [87,88].

Platelets are major players in the NO/sGC/PKGI pathway, likely through final substrates such as VASP and IRAG [21]. PKGI activation and the phosphorylation of VASP and IRAG appear to inhibit platelet aggregation [21]. Vuorinen et al. [89] also reported that cGMP inhibits thrombin-induced aggregation and serotonin secretion. Furthermore, cGMP-dependent mechanisms inhibit platelet apoptosis, reducing platelet activation and aggregation [90].

In vitro studies [91] indicate that sGC activation in platelets influences atherosclerotic plaque formation and vascular inflammation. Mauersberger et al. [91] found that sGC inhibition reduces sGC-dependent platelet-derived angiopoietin-1 release, limiting leukocyte adhesion to endothelial cells.

### 3.4. Future Perspectives

Although the NO-sGC-cGMP pathway shows strong potential in vascular health, further studies are needed to confirm its effects across different cell types and disease conditions. Key areas of interest include the context-dependent role of BKCa channels in VSMCs, the balance between sGC stimulators and activators in endothelial function, and the emerging impact of sGC activity in bloodstream cells like RBCs and platelets.

Future clinical research should focus on translating these findings into therapies for cardiovascular diseases, especially where oxidative stress and endothelial dysfunction are prominent.

## 4. Kidneys and Renal Function

The impact of cardiovascular disease—particularly HF—on kidney function is well-established [92]. The reciprocal interaction between the heart and kidneys is crucial for evaluating prognosis and the quality of life of patients, as it is well known that both conditions exponentially increase the risk for these patients [93].

Regardless of the underlying etiology of kidney pathologies, it has been demonstrated that the disruption of the NO/sGC/PKGI pathways impacts the occurrence and progression of kidney failure, as well as HF (Figure 3) [94].

The glomerulus as a whole is a key target for the NO/sGC/PKGI effectors. The biochemical pathways associated with this axis influence vascular tone, cellular proliferation, inflammation, and fibrosis, thereby regulating kidney function and contributing to dysfunction in cases of alterations [94]. The expression of sGC in the renin-producing granular cells, mesangium, descending vasa recta, and cortical and medullary interstitial fibroblasts supports these findings [95].

Different effects have been observed on the vascular tone of the efferent and afferent arterioles of the glomerulus. Stimulators of sGC appear capable of dilating both arterioles and improving renal blood flow under oxidative stress conditions [96]. Specifically, Wennysia et al. [97] demonstrated more pronounced vasodilation of the efferent arterioles compared to the afferent arterioles when sGC stimulators were used in the preclinical setting. Although renal blood flow autoregulation and the renovascular myogenic response play a role in controlling the vascular tone of the glomerular vessels, sGC activity is essential for maintaining proper kidney perfusion [98]. This is further supported by the role of sGC in preserving the integrity of podocytes, thus protecting the glomerulus from focal segmental sclerosis and functional impairment [99]. The stimulation of sGC appears to reduce the expression and activity of the Ca^2+^ channel TRPC6, which is primarily linked to glomerulosclerosis [99]. Preclinical studies [100] found that stimulation of the sGC-cGMP-PKG pathways in renal and aortic structures reduced vascular calcification, renal fibrosis, and endothelial dysfunction. Such regulation may be disrupted during ischemic injuries: ischemia/reperfusion injuries can reduce sGC activity and lower cGMP concentrations, contributing to insufficient renal perfusion and worsened renal function [101]. The activation of soluble guanylyl cyclase signaling with cinaciguat was shown to reduce the activation of Thrombospondin 1 [102], which is a key promoter of fibroproliferative pathways in the kidneys [103]. This helps explain the reduction in glomerulosclerosis, interstitial, and perivascular fibrosis of intrarenal arteries, demonstrated after the stimulation of the sGC-cGMP-PKG pathway via cinaciguat [104]. Schinner et al. [105] found that sGC BAY 41-8543 inhibited the TGFβ signaling pathway, decreased the phosphorylation of Erk 1 and 2, and reduced the activation of downstream mediators such as Smad2 and Smad3. This translated into a reduction in the activity of extracellular matrix (ECM)-degrading matrix metalloproteases (MMP2 and MMP9) and the promotion of metalloproteinase inhibitors-1 (TIMP-1) [105]. The final result was the protection of the kidneys from fibrosis. Similarly, Shea et al. [106] confirmed the positive effects of praliciguat—another sGC stimulator—on renal structure and function. The administration of praliciguat (10 mg·kg^−1^·day^−1^) to Dahl salt-sensitive rats fed a high-salt diet significantly reduced the expression of genes such as tissue inhibitor of metallopeptidase 1, collagen type III-α1, interleukin-6, NF-kB subunit 1, TNF-α, monocyte chemoattractant protein-1, Vascular Cell Adhesion Molecule 1, Intercellular Adhesion Molecule-1, and *Tgfb1* [106]. Histological evaluations of the kidneys confirmed the reduction in glomerulosclerosis, interstitial fibrosis, interstitial inflammation, and vascular alterations in those rats treated with praliciguat [106].

The action of the sGC-cGMP-PKG pathway on the kidney also involves various parts of the glomerulus. Preclinical [107] and clinical [108] studies demonstrated that the NO/cGMP axis might counteract the action of angiotensin II on sodium retention in the proximal tubules by affecting ion transporters.

Specifically, preclinical studies observed that the activity of the inwardly rectifying K^+^ channel in cultured human proximal tubule cells could be influenced by a GMP/PKG-dependent mechanism, with higher nitric oxide concentrations impairing the activation of this channel [109]. The inwardly rectifying K^+^ channel is known to regulate Na^+^, K^+^, and bicarbonate transport in the proximal tubules by interacting with the electrogenic Na^+^-bicarbonate cotransporter and the type 3 Na^+^/H^+^ exchanger (NHE3) [110]. Furthermore, the sGC-cGMP axis may also regulate the activity of angiotensin (AT) receptors. Activation of the AT-2 receptor can inhibit the AT-1 receptor’s function on Na^+^/K^+^-ATPase in the renal proximal tubules via the G(i/o) protein/cGMP/PKG pathway, thus altering reabsorption in this part of the glomerulus [111,112].

These findings were supported by Newaz et al., who demonstrated that the generation of NO activated Peroxisome Proliferator-Activated Receptor-α, thereby increasing renal Na^+^ excretion due to reduced activity of Na^+^/K^+^-ATPase [113].

The activation of sGC in proximal tubules contributes to the maintenance of their integrity and performance by reducing inflammatory conditions, such as monocyte chemoattractant protein-1 secretion, and apoptosis via the reduction in TGF-β expression [114].

The sGC-cGMP axis also affects distal tubules and collecting ducts. Garcia et al. [115] observed the natriuretic effects of NO in promoting the inhibition of osmotic water permeability due to the action of antidiuretic hormone in this section of the glomerulus. NO appears to activate a cGMP-dependent protein kinase, which, in turn, inhibits antidiuretic hormone-stimulated cyclic adenosine monophosphate production [115]. The natriuretic effects of the NO-sGC axis may also be mediated by its influence on aquaporins (APs). While natriuretic peptides have a well established role in modulating the translocation of AP2 from the cytoplasm to the membrane of collecting duct cells [116,117], the influence of the sGC-cGMP axis should also be considered. Klokkers et al. [118] found a reduction in cGMP that led to decreased AP2 translocation from the cytoplasm to the membranes, thus preventing water retention associated with AP2. However, further research is needed to confirm these findings.

Therefore, the NO-sGC-cGMP pathway shows promising renal protective effects by improving glomerular vascular tone, reducing fibrosis, and modulating ion transport in tubular cells. Understanding how this pathway interacts with sodium handling and aquaporin regulation could open new therapeutic strategies for preventing renal fibrosis, glomerulosclerosis, and progressive renal dysfunction.

## 5. Conclusions

The NO-sGC-cGMP axis has a wide range of beneficial effects on cardiovascular and renal function. The stimulation of sGC effectively enhances the performance of cardiac muscle cells and the cardiac environment.

Furthermore, positive effects on endothelial function and the vascular system are mediated by influencing the activity, structure, and inflammatory status of vascular smooth muscle cells, endothelial cells, and blood cells.

Finally, the biochemical pathways related to sGC seem to improve kidney function by influencing the morphology and activity of various parts of the glomerulus: they maintain capillary relaxation (in vessels both within and outside the macula densa), preserve the integrity of kidney structures (tubules, macula densa, and podocytes), and reduce interstitial fibrosis as well as mesangial cell contraction and expansion due to inflammation.

Thus, sGC stimulators represent a promising therapeutic approach to counteract the loss of cardiac, vascular, and kidney structures. Further research is needed to validate these findings.

## Figures and Tables

**Figure 1 ijms-26-04550-f001:**
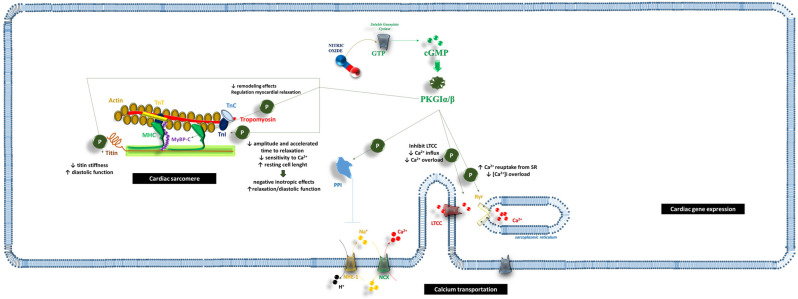
The impact of NO/sGC/cGMP on cardiac cells. The biochemical pathway related to soluble guanylate cyclase (sGC) activation influences the performance of cardiac sarcomere, ion transportation throughout the cardiac muscular cells, and the expression of genes which are crucial for survival, cell growth, and ultrastructural modification of the cell. Abbreviations: Ca^2+^: calcium ion; cGMP: cyclic guanosine monophosphate; GATA4: GATA Binding Protein 4; GTP: guanosine-5′-triphosphate; H^+^: hydrogen ion; LTCC: L-type calcium channel; MAPK: mitogen-activated protein kinase; MEF2: Myocyte Enhancer Factor 2; MHC: myosin heavy chains; mTROC1: mammalian target of rapamycin complex 1; MyBP-C: myosin-binding protein C; Na^+^: sodium ion; NCX: sodium–calcium exchanger; NHE-1: sodium–hydrogen antiporter 1; NO: nitric oxide; PKGIα/β: cGMP-dependent protein kinase I α/β; PPI: protein phosphatase 1; Ryr: ryanodine receptor; sGC: soluble guanylate cyclase; SR: sarcoplasmic reticulum; TAB: TGF-Beta-Activated Kinase; TnC: troponin C; TnI: troponin I; TnT: troponin T; TSC2: tuberous sclerosis complex 2.

**Figure 2 ijms-26-04550-f002:**
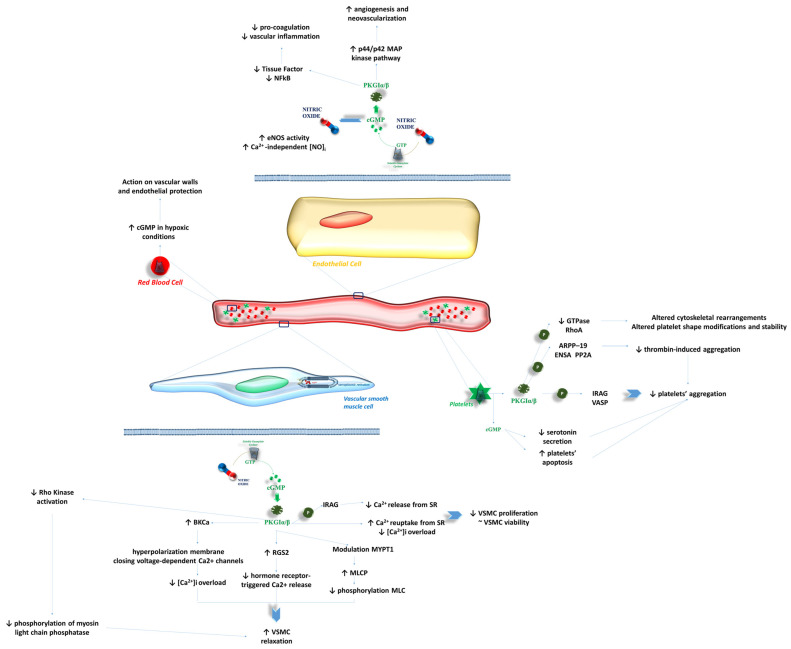
The influence of NO/sGC/cGMP activity on peripheral vascular wall morphology and function. The activation of the soluble guanylate cyclase on peripheral vascular cells might influence the structure and function of endothelial cells, vascular smooth muscle cells, and circulating red blood cells/platelets. Abbreviations: ARPP-19: cAMP-regulated phosphoprotein 19; BKCa: large-conductance Ca^2+^-activated K^+^ channel; Ca^2+^: calcium ion; cGMP: cyclic guanosine monophosphate; ENSA: endosulfine alpha; GTP: guanosine-5′-triphosphate; GTPase: guanosine-5′-triphosphate hydrolase; IRAG: inositol-1,4,5-trisphosphate receptor-associated cGMP kinase substrate; MAP kinase: mitogen-activated protein kinase; MLC: Myosin-light-chain; MLCP: myosin-light-chain phosphatase; MYPT1: myosin phosphatase target subunit 1; NF-kB: nuclear factor kappa-light-chain-enhancer of activated B cells; NO: nitric oxide; eNOS: endothelial nitric oxide synthase; PKGIα/β: cGMP-dependent protein kinase I α/β; PPA2: Protein Phosphatase 2A; RGS2: regulator of G-protein signaling 2; sGC: particulate guanylate cyclase; SR: sarcoplasmic reticulum; VASP: vasodilator-stimulated phosphoprotein; VSMC: vascular smooth muscle cells.

**Figure 3 ijms-26-04550-f003:**
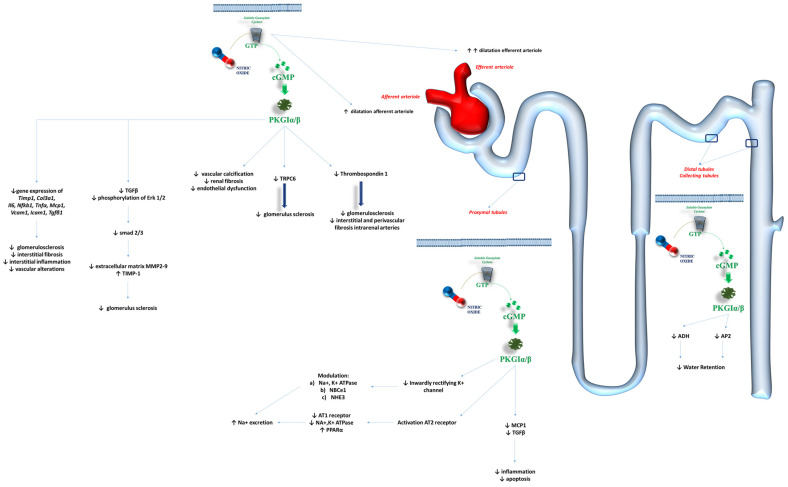
The role of the NO/sGC/cGMP biochemical pathway on the glomerulus activity and structure. The biochemical pathway related to the activation of the soluble guanylate cyclase might impact the structure and function of each part of the glomerulus, thus promoting antiproliferative and antifibrotic remodeling, modification in ultrafiltration skills, and reduction in inflammatory degeneration. Abbreviations: ADH: antidiuretic hormone; AP2: aquaporin 2; AT1: angiotensin 1 receptor; AT2: angiotensin 2 receptor; cGMP: cyclic guanosine monophosphate; Col3a1: collagen type III alpha 1 chain; ERK: extracellular signal-regulated kinases; GTP: cuanosine-5′-triphosphate; Icam1: Intercellular Adhesion Molecule 1; Il6: inteleukin-6; K^+^: potassium ion; MCP1: monocyte chemoattractant protein-1; MMP2-9: matrix metalloproteinase-2 and -9; Na^+^: sodium ion; NBCe1: electrogenic sodium bicarbonate cotransporter 1; NF-kB: nuclear factor kappa-light-chain-enhancer of activated B cells; NHE-1: sodium–hydrogen antiporter 1; NO: nitric oxide; PKGIα/β: cGMP-dependent protein kinase I α/β; PPARα: Peroxisome Proliferator-Activated Receptor α; sGC: particulate guanylate cyclase; smad: small mother against decapentaplegic; TGFβ: transforming growth factor-β; TIMP: tissue inhibitor of metalloproteinase; Tnfα: tumor necrosis factor-α; TRPC6: transient receptor potential cation channel subfamily C member 6; Vcam1: vascular cell adhesion protein 1.

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
