# Peer review of "Influence of Soluble Guanylate Cyclase on Cardiac, Vascular, and Renal Structure and Function: A Physiopathological Insight"

_ijms, 2025, doi:10.3390/ijms26104550_

Round 1
Reviewer 1 Report
Comments and Suggestions for Authors
The manuscript entitled "Guanylate cyclase/cGMP axis and cardiovascular/renal structure and function: a physiopathological insight" provides a comprehensive narrative review of the nitric oxide (NO)–soluble guanylate cyclase (sGC)–cyclic GMP (cGMP) pathway and its relevance to cardiovascular and renal physiology and pathophysiology. The topic is timely and relevant, especially given the recent clinical applications of sGC stimulators such as vericiguat. The authors aim to synthesize current knowledge regarding the pleiotropic effects of the NO–sGC–cGMP axis on cardiac muscle cells, vascular function, and kidney physiology. Overall, the manuscript is well-structured and addresses an important area in molecular cardiology and nephrology. However, several conceptual clarifications, terminology corrections, and improvements in the interpretation of key mechanisms are needed to enhance the scientific rigor and clarity of the review. My specific comments are detailed below.
Major Points
- Figure 1 contains a conceptual error. Nitric oxide (NO) does not activate particulate guanylate cyclase (pGC). pGC is specifically activated by natriuretic peptides (such as ANP, BNP, and CNP) and is not responsive to NO. NO exclusively activates soluble guanylate cyclase (sGC), a cytosolic, heme-containing enzyme that is sensitive to direct NO binding.
- Throughout the manuscript, the authors often conflate the terms “soluble guanylate cyclase” (sGC) and “particulate guanylate cyclase” (pGC)—particularly in figure legends and abbreviations. sGC is cytosolic and NO-activated, while pGC is membrane-bound and activated by natriuretic peptides. The authors should revise all mentions of “sGC: particulate guanylate cyclase” and accurately distinguish between these enzymes. This issue should also be corrected in the figures.
- Line 174. “In another study, Fiedler et al. [45] demonstrated that NO and its second messenger, cGMP, protect cardiomyocytes from apoptosis during ischemia-reperfusion. cGMP activates PKGI, which interferes with the TAB/p38 mitogen-activated protein kinase (p38 MAPK) pathway, a key regulator of apoptosis [45].”
→ The authors should provide a more detailed discussion of how this pathway is modulated. - The text mentions protection against ischemia-reperfusion injury but does not highlight the direct effects of the cGMP–PKG pathway on mitochondrial function—such as the inhibition of mitochondrial permeability transition pore (mPTP) opening.
- The role of TRPC channels is correctly mentioned, but it is not the only pathway involved in Ca²⁺ influx in cardiomyocytes. The authors should clarify that this is “one of the involved pathways” to avoid misinterpretation by readers.
- Section 3.2. The authors state that sGC can induce NO production by activating NOS. However, literature suggests that cGMP may modulate NOS activity, but it is not accurate to say that sGC directly “stimulates” NOS. The authors should better explain this mechanism, particularly regarding eNOS activation. A more appropriate phrasing would be: “cGMP may indirectly enhance NOS activity in endothelial cells through feedback mechanisms.”
- BAY 41-2272 and BAY 58-2667 do not belong to the same pharmacological class. There is an important distinction between sGC stimulators and sGC activators. The authors should explain the difference in their mechanisms of action.
- In the cited study by Schmid et al., NO is shown to reduce BKCa activity. This contradicts the majority of the literature, which associates NO–cGMP–PKG signaling with activation of BKCa channels. The authors should contextualize this finding and clarify that BKCa involvement in NO-induced vasodilation may be vessel- and context-dependent.
- The authors should clearly indicate which studies were conducted in humans and which in animals.
- In several instances, the authors cite review articles instead of original research. Whenever possible, they should reference the original articles to allow readers to verify findings or consult the full study.
- The authors should describe the contents of Figures 1, 2, and 3 more explicitly within the main text.
- The authors should clarify whether the calcium-dependent NOS isoform mentioned refers to eNOS.
- In Figure 2, “calcium-independent NOS” appears twice. One of them likely refers to calcium-dependent NOS and should be corrected.
- The authors should include a perspectives/future directions paragraph at the end of each major section.
- The discussion on sGC stimulators and activators should be expanded, particularly focusing on their differing effects in oxidized versus reduced forms of sGC.
- The title should be revised in light of the above corrections, as the paper primarily focuses on sGC-related mechanisms.
Minor Points
- Standardize nomenclature for ion channels, proteins, and receptors throughout the text (e.g., capitalization, Greek letters, subunit formatting).
- Abbreviations are defined multiple times. Consider creating a consolidated list of abbreviations at the beginning or end of the manuscript.
Author Response
We thank this Reviewer for the constructive comments and suggestions. Furthermore, we would like to really thank him/her for his/her appreciation about our research. This is our point to point reply.
The manuscript entitled "Guanylate cyclase/cGMP axis and cardiovascular/renal structure and function: a physiopathological insight" provides a comprehensive narrative review of the nitric oxide (NO)–soluble guanylate cyclase (sGC)–cyclic GMP (cGMP) pathway and its relevance to cardiovascular and renal physiology and pathophysiology. The topic is timely and relevant, especially given the recent clinical applications of sGC stimulators such as vericiguat. The authors aim to synthesize current knowledge regarding the pleiotropic effects of the NO–sGC–cGMP axis on cardiac muscle cells, vascular function, and kidney physiology. Overall, the manuscript is well-structured and addresses an important area in molecular cardiology and nephrology. However, several conceptual clarifications, terminology corrections, and improvements in the interpretation of key mechanisms are needed to enhance the scientific rigor and clarity of the review. My specific comments are detailed below.
Major Points
- Figure 1 contains a conceptual error. Nitric oxide (NO) does not activate particulate guanylate cyclase (pGC). pGC is specifically activated by natriuretic peptides (such as ANP, BNP, and CNP) and is not responsive to NO. NO exclusively activates soluble guanylate cyclase (sGC), a cytosolic, heme-containing enzyme that is sensitive to direct NO binding.
We would like to really thank the reviewer for this comment and further apologize for the mistake. Despite the revisions of the paper we thoroughly continued to be wrong with the figures. Thanks so much for this evaluation. We revise ALL the figures in order to amend the wrong representation.
- Throughout the manuscript, the authors often conflate the terms “soluble guanylate cyclase” (sGC) and “particulate guanylate cyclase” (pGC)—particularly in figure legends and abbreviations. sGC is cytosolic and NO-activated, while pGC is membrane-bound and activated by natriuretic peptides. The authors should revise all mentions of “sGC: particulate guanylate cyclase” and accurately distinguish between these enzymes. This issue should also be corrected in the figures.
Once again thank you very much for this comment. As previously outlined, we revised the entire manuscript and figures in order to amend this wrong.
- Line 174. “In another study, Fiedler et al. [45] demonstrated that NO and its second messenger, cGMP, protect cardiomyocytes from apoptosis during ischemia-reperfusion. cGMP activates PKGI, which interferes with the TAB/p38 mitogen-activated protein kinase (p38 MAPK) pathway, a key regulator of apoptosis [45].” → The authors should provide a more detailed discussion of how this pathway is modulated.
Thank you very much for this insight. We amend the text in order to make it clearer.
- The text mentions protection against ischemia-reperfusion injury but does not highlight the direct effects of the cGMP–PKG pathway on mitochondrial function—such as the inhibition of mitochondrial permeability transition pore (mPTP) opening.
Thanks for this comment. We include a dedicated paragraph dealing with the impact of cGMP-PKG pathway on mitochondrial function.
- The role of TRPC channels is correctly mentioned, but it is not the only pathway involved in Ca²⁺ influx in cardiomyocytes. The authors should clarify that this is “one of the involved pathways” to avoid misinterpretation by readers.
Thank you very much for this comment. We effectively amend the text in order to comply with this issue.
- Section 3.2. The authors state that sGC can induce NO production by activating NOS. However, literature suggests that cGMP may modulate NOS activity, but it is not accurate to say that sGC directly “stimulates” NOS. The authors should better explain this mechanism, particularly regarding eNOS activation. A more appropriate phrasing would be: “cGMP may indirectly enhance NOS activity in endothelial cells through feedback mechanisms.”
Thank you for this feedback. We accordingly revised the text.
- BAY 41-2272 and BAY 58-2667 do not belong to the same pharmacological class. There is an important distinction between sGC stimulators and sGC activators. The authors should explain the difference in their mechanisms of action.
Once again we would like to really thank the reviewer for this interesting insight. We amend the text in order to outline the differences between the two compounds.
- In the cited study by Schmid et al., NO is shown to reduce BKCa activity. This contradicts the majority of the literature, which associates NO–cGMP–PKG signaling with activation of BKCa channels. The authors should contextualize this finding and clarify that BKCa involvement in NO-induced vasodilation may be vessel- and context-dependent.
We do agree with the reviewer. We revised the paper in relation to the comments.
- The authors should clearly indicate which studies were conducted in humans and which in animals.
We can understand the concern from the reviewer. Indeed, the entire manuscript is fundamentally based on preclinical studies. In vitro/in vivo studies and the adoption of animals models represent the totality of the references. We considered a few clinical studies and we rather specify when clinical studies and human involvement were included.
- In several instances, the authors cite review articles instead of original research. Whenever possible, they should reference the original articles to allow readers to verify findings or consult the full study.
Thank you for the suggestion. We implemented references by including original research in agreement with the review articles that were already included in the manuscript.
- The authors should describe the contents of Figures 1, 2, and 3 more explicitly within the main text.
Thanks for this comment. Indeed, we avoided the description of each figures for two reasons: the first was related to length issues. We preferred not to prolong the contents of the manuscript. Secondly, the figures are a representation of the contents in each paragraph of the text. They summarized via an iconographic manner the information and results of the analysis of the literature.
- The authors should clarify whether the calcium-dependent NOS isoform mentioned refers to eNOS.
We thank the reviewer for this comment. The NOS isoform was the eNOS. We specified it in the main text.
- In Figure 2, “calcium-independent NOS” appears twice. One of them likely refers to calcium-dependent NOS and should be corrected.
We amend figure 2 in order to make it clearer. Thanks for the comment.
- The authors should include a perspectives/future directions paragraph at the end of each major section.
We included a dedicated paragraph on perspectives/future directions at the end of each major section.
- The discussion on sGC stimulators and activators should be expanded, particularly focusing on their differing effects in oxidized versus reduced forms of sGC.
We would like to really thank the reviewer for this further, effectively interesting insight. We have added a paragraph dealing with this issue.
- The title should be revised in light of the above corrections, as the paper primarily focuses on sGC-related mechanisms.
Thank you for this comment. We revise the title.
Minor Points
- Standardize nomenclature for ion channels, proteins, and receptors throughout the text (e.g., capitalization, Greek letters, subunit formatting).
We tried to standardize the nomenclature as indicated. Thanks for the suggestion.
- Abbreviations are defined multiple times. Consider creating a consolidated list of abbreviations at the beginning or end of the manuscript.
We included a list of abbreviations at the end of the manuscript.
Reviewer 2 Report
Comments and Suggestions for Authors
The submitted paper is an attempt to review the role of the intracellular cGMP pathway in heart and kidney under different pathological conditions. This aim is well reasoned as several drugs in clinical use and being under development target that pathway, and do it at the same time in different organs, tissues and cell types.
The title names the topic as guanylate cyclase, this is declared later (in the introduction and elsewhere) to be reduced to the soluble (cytoplasmic) form of the enzyme However, in all the Figures unanimously the membrane-bound form of the enzyme is visualized. A substantial part of the paper deals with endothelium and vascular smooth muscle, not covered by the title. An other essential problem is that the activity of the cGMP pathway is heavily dependent on previous steps of activation: NO production in endothelial cells (shear stress, aging, diabetes, hypertensive wall damage, arteriosclerosis, several physiological and pharmacological agonists, inflammatory remodeling of endothelial cells, etc.); their discussion in different cell types and in different pathologies can hardly be avoided to provide a systematic review. An important agonist with cGMP signal pathway is atrial natriuretic peptide (ANP). Its discussion is almost fully missing from the paper despite its substantial role in heart failure and several kidney pathologies.
There are many inconsistencies in the otherwise aesthetic and valuable Figures. Just a few of them: On all Figures membrane-bound guanylate cyclase is shown, soluble guanylate cyclase does not appear at all. NO is shown to attack the membrane bound guanylate cyclase only from the interior of the cell. The cGMP is marked as 3 molecules attaching to some protein structure (?). On Fig 1. cGMP effects are partly through PKG, partly directly (?). From Fig 2. (smooth muscle relaxation effect) the steps of inhibition of RhoA-Rhokinase are missing (Kato M et al, J Biol Chem 2012;287:41342). Rho kinase has several important additional functions). In case of distal nephron (Fig.3) ANP is not even mentioned. All Figures should be step-by-step checked by an expert with substantial overview of the full topic. Structural pathologic changes are stressed at many places, such will be only very late effects, and hardly the result of alteration in the cGMP signal pathway only.
Author Response
We thank this Reviewer for her/his useful suggestions. We sincerely appreciate his/her comments on our work. This is our point-to-point reply:
The submitted paper is an attempt to review the role of the intracellular cGMP pathway in heart and kidney under different pathological conditions. This aim is well reasoned as several drugs in clinical use and being under development target that pathway, and do it at the same time in different organs, tissues and cell types.
- The title names the topic as guanylate cyclase, this is declared later (in the introduction and elsewhere) to be reduced to the soluble (cytoplasmic) form of the enzyme However, in all the Figures unanimously the membrane-bound form of the enzyme is visualized.
We are really sorry for this fundamental wrong in the figures. We amend the figures in order to comply to the real focus of the paper.
- A substantial part of the paper deals with endothelium and vascular smooth muscle, not covered by the title.
Thank you for this comment. We revise the title.
- Another essential problem is that the activity of the cGMP pathway is heavily dependent on previous steps of activation: NO production in endothelial cells (shear stress, aging, diabetes, hypertensive wall damage, arteriosclerosis, several physiological and pharmacological agonists, inflammatory remodeling of endothelial cells, etc.); their discussion in different cell types and in different pathologies can hardly be avoided to provide a systematic review.
We do agree with this comment. Nevertheless, we tried to mainly outline and discuss the role of guanylate cyclase in this setting and we intentionally avoided the discussion about the role of NO and the factors that influence NO production and concentration. A further discussion about this issue would increase the length of paper and go beyond the aims of this narrative review, but it will be a great starting point for the development of further insights and manuscripts.
- An important agonist with cGMP signal pathway is atrial natriuretic peptide (ANP). Its discussion is almost fully missing from the paper despite its substantial role in heart failure and several kidney pathologies.
We would like to really thank the reviewer for this insight. Natriuretic peptides effectively impact on heart function and dysfunction as well as on kidneys. Indeed, we voluntary did not mention the role of natriuretic peptide on cGMP signal pathway. As the main aim of this narrative review was to provide a pathophysiologic insight on soluble guanylate cyclase, we preferentially avoid dealing with particulate GC and the influence of natriuretic peptides on this receptor.
- There are many inconsistencies in the otherwise aesthetic and valuable Just a few of them:
- On all Figures membrane-bound guanylate cyclase is shown, soluble guanylate cyclase does not appear at all. NO is shown to attack the membrane bound guanylate cyclase only from the interior of the cell.
We really apologize for this mistake. We amend the figures in relation to the aims and the contents of the paper.
- The cGMP is marked as 3 molecules attaching to some protein structure (?).
Thank you for this comment. Indeed, the 3 molecules are 3 cGMP different molecules. It is not a stechiometric representation of the number of cGMP molecules produced by the activation of sGC, but rather a rough representation of an “indefinite” number of cGMP molecules that might be produced after the activation of sGC.
- On Fig 1. cGMP effects are partly through PKG, partly directly (?).
The effects of cGMP are mediated by PKG as outlined in the main text.
- From Fig 2. (smooth muscle relaxation effect) the steps of inhibition of RhoA-Rhokinase are missing (Kato M et al, J Biol Chem 2012;287:41342). Rho kinase has several important additional functions).
Thank you very much for the suggestion. We amend the figure 2 in relation to this comment.
- In case of distal nephron (Fig.3) ANP is not even mentioned.
Thank you for this comment. Indeed, as previously mentioned, we did not consider the role of ANP in this setting as the main aim of the manuscript was related to sGC rather than particulate GC. For this reason, we did not consider and discuss the role of natriuretic peptides in the context of this narrative review.
- All Figures should be step-by-step checked by an expert with substantial overview of the full topic. Structural pathologic changes are stressed at many places, such will be only very late effects, and hardly the result of alteration in the cGMP signal pathway only.
Thank you for this comment. Figures derived from the analysis of the text and the literature. We schematically included in each figures all the information which were included in the text and derived from a previous analysis of the literature.
Round 2
Reviewer 1 Report
Comments and Suggestions for Authors
I have no question
Reviewer 2 Report
Comments and Suggestions for Authors
Additions, alterations in the revised form much improved the paper, now it is a valuable, almost full overview of this clinically and theoretically important topic.